The impact of new package managers on the library dependency ecosystem

Rahkema Kristiina 1 kristiina.ra@gmail.com
http://orcid.org/0000-0003-2400-501X Pfahl Dietmar 1
http://orcid.org/0000-0001-9903-6107 Ramler Rudolf 2
1 Institute of Computer Science, University of Tartu , Tartu , Estonia
2 Software Competence Center Hagenberg , Hagenberg , Austria
Graziotin Daniel
Electronic publication date: 2024 Dec 20
Publication date: 2024
Volume: 10
Electronic Location ID: e2617
Received 2024 Feb 9; Accepted 2024 Nov 27
Copyright: © 2024 Rahkema et al.
Copyright year: 2024
Copyright holder: Rahkema et al.
License: This is an open access article distributed under the terms of the Creative Commons Attribution License, which permits unrestricted use, distribution, reproduction and adaptation in any medium and for any purpose provided that it is properly attributed. For attribution, the original author(s), title, publication source (PeerJ Computer Science) and either DOI or URL of the article must be cited.
License URL: https://creativecommons.org/licenses/by/4.0/

Keywords: iOS, Library dependencies, Package managers, Evolution

Funding: BMK, BMAW, and the State of Upper Austria SCCH Competence Center INTEGRATE 892418 European Regional Development Fund PRG1226 The research reported in this article has been funded by BMK, BMAW, and the State of Upper Austria in the frame of the SCCH Competence Center INTEGRATE (FFG grant no. 892418) part of the FFG COMET Competence Centers for Excellent Technologies Programme, as well as by the European Regional Development Fund, and grant PRG1226 of the Estonian Research Council. There was no additional external funding received for this study. The funders had no role in study design, data collection and analysis, decision to publish, or preparation of the manuscript.

==============================
Adding dependencies to third-party libraries through package managers is a common practice in software development. The evolution of library dependency networks has been analyzed for many package managers. There are, however, no studies on how the library dependency networks of multiple package managers behave in the same ecosystem. The library dependency network in the Swift ecosystem encompasses libraries from CocoaPods, Carthage, and Swift Package Manager (Swift PM). These three package managers are used when developing, for example, iOS or macOS applications in Swift or Objective-C. In this study, we analyze how the introduction of new package managers has affected the evolution of the library dependency network of the Swift ecosystem. We found that overall the popularity of using package managers has grown over time. We saw that the introduction of Carthage and Swift PM had some but not a large influence on the popularity of CocoaPods. Carthage users; however, are increasingly migrating to Swift PM. This discrepancy could stem from the fundamental differences between CocoaPods and the other two package managers, as well as similarities between Carthage and Swift PM. Based on our observations, we speculate that Apple could increase the popularity of Swift PM by adding features that have so far only been available in CocoaPods, such as a central repository.

Introduction

Using third-party libraries is common practice in software development. Third-party libraries allow developers to reuse existing solutions to common problems, which can make the development process faster and easier.

The libraries themselves can have dependencies on other libraries, creating a potentially complex network of library dependencies. Using a package manager is a convenient way to handle these dependencies by simply declaring which dependencies on third-party libraries exist in a project. The package manager resolves all dependencies and includes the necessary libraries. The collection of all libraries that are available through a package manager and their library dependencies creates a library dependency network (LDN) for each package manager.

There is a package manager for almost every ecosystem, for example Maven for Java and npm for JavaScript. In some ecosystems, developers can even choose between multiple package managers. In the Swift ecosystem, for example, three package managers are available: CocoaPods, Carthage, and Swift Package Manager (Swift PM). These package managers are used when developing iOS or macOS applications in Swift or Objective-C. Package managers in the Swift ecosystem were introduced over time, CocoaPods in 2011 and Carthage in 2014. Swift PM is the official package manager by Apple that was most recently introduced in 2017. No clear guidelines exist for developers on which package manager should be chosen for a new project. On a superficial level CocoaPods seems to provide access to most libraries, but this might be the case simply due to CocoaPods being the only package manager with a central repository, making all available libraries directly visible to users. A thorough analysis of the three LDNs might help make the package manager ecosystem more clear for iOS and macOS developers.

In this study, we analyze how the introduction of new package managers has affected the evolution of the library dependency ecosystem. The release of a new package manager might cause users to migrate from an older package manager. The number of libraries in dependency networks may grow. Developers not using any package manager may start using one for managing dependencies on third-party libraries. Such insights are important for developers of third-party libraries, for tool developers, as well as for their users.

Existing research, so far, has analyzed each package manager ecosystem separately. The LDNs of many package managers have been studied and compared. For example, Kikas et al. (2017) created a dependency dataset and studied the LDNs of JavaScript, Ruby and Rust. Decan, Mens & Grosjean (2019) used the libraries.io dataset to study the growth of LDNs of seven package managers. No study so far, however, has analyzed multiple package managers co-existing in the same ecosystem.

Several ecosystems exist where multiple package managers are available. Sometimes a non-official package manager is released that serves a niche subset of the ecosystem (e.g., Bioconductor (https://www.bioconductor.org) for Python that is used for bioinformatics-related libraries). Sometimes a new package manager is released, but the underlying library repository stays the same (for example, npm and Yarn for JavaScript (Yarn, 2023)). In other cases, each package manager has its own separate library dependency network.

We choose to analyze the Swift ecosystem encompassing the three popular and actively used package managers CocoaPods, Carthage, and Swift PM. The Swift ecosystem contains libraries used in iOS and macOS development. It is a large and, to some extent, closed ecosystem with a long history and a strong user base. The three package managers have been introduced over the lifetime of the ecosystem, allowing us to analyze how the introduction of each of the package managers influenced the library dependency network of the entire ecosystem.

In our study, we investigate three research questions, asking (1) how the overall popularity of package managers has evolved, (2) if the three package managers are used concurrently, and (3) how the introduction of new package managers influences the evolution of the ecosystem.

We found that overall the popularity of using package managers has grown over time. We saw that the introduction of Carthage and Swift PM had some, but not a large, influence on the popularity of CocoaPods. Carthage users, however, are increasingly migrating to Swift PM. This discrepancy could stem from the fundamental differences between CocoaPods and the other two package managers, as well as similarities between Carthage and Swift PM. Apple has made efforts to facilitate the use of package managers by incorporating Swift PM seamlessly into Xcode, the main integrated development environment (IDE) used in iOS and macOS development, increasing its adaption by developers. Based on our observations, we speculate that Apple could increase the popularity of Swift PM by adding features that have so far only been available in CocoaPods, such as a central repository.

In the context of the iOS library dependency ecosystem, the contribution of our research can be summarized as follows: We provide insights into how the popularity of package managers has changed over time.

We explore how package managers are used concurrently and provide insights into how this changes over time.

We provide the first results of an analysis on how the release of new package managers affects the evolution of the package manager ecosystem-in the iOS context and beyond.

We discuss properties of package managers that may increase their acceptance by developers.

The rest of the article is structured as follows. In the Background Section, we describe the iOS/macOS ecosystem, the three package managers, how dependencies are declared for each package manager, and give an overview of the dataset used. In the Method Section, we describe the research questions and how we analyzed the dataset for each research question. In ‘Results’, we present answers to the three research questions and ‘Discussion’ discusses these answers. In ‘Threats to Validity’, we describe the threats to validity. In ‘Related Work’, we summarize related work and, in the ‘Conclusion’, we discuss promising improvements of package managers and conclude the article.

Portions of this text were previously published as part of a PhD thesis by Rahkema (2023) and a preprint by Rahkema & Pfahl (2023) of an earlier version of this article.

Background

This section describes the iOS/macOS ecosystem, the selected package managers, and the dataset we used in our analyses.

The iOS/macOS ecosystem

Apple has created a fairly unique ecosystem where software and the operating systems are running on the hardware developed by the same company. The wide selection of Apple devices ranging from personal computers to smartphones and smart watches provides opportunities for the optimisation of hardware and software and the interoperability between different devices. This unique closedness of the ecosystem (as opposed to for example the openness of the Android ecosystem) makes it an interesting subject to study as it allows to trace the evolution over time with only minimal influences from other ecosystems.

Figure 1 gives an overview of some of the milestones in the evolution of the macOS/iOS ecosystem. The first Mac OS X version (10.0) was released already in 2001. The X in the name indicated that the operating system was a Unix system. The Macintosh computers built by Apple at the time were based on PowerPC. In 2003, Apple released the first Xcode version to improve the development of Mac OS X applications.

Figure 1 Timeline of the macOS/iOS ecosystem.

The figure shows a timeline of the macOS/iOS ecosystem highlighting some important events. The red area shows the number of apps in the iOS app store over time. Events with red titles show important events in the iOS ecosystem, events with green titles show important events in the macOS ecosystem and events with blue titles show important events related to package managers.

In 2007, Apple released the first iPhone and its operating system iPhone OS 1. A year later the App Store was launched as distribution platform for iOS applications. The red area in Fig. 1 shows the growth of the App Store as the number of iOS applications.

Following in 2011, the Mac App Store was launched for macOS applications. Mac OS X was renamed to OS X in 2012 with the release of OS X 10.8. Later, in 2016, it was renamed again to macOS with the release of macOS 10.12.

The official language for developing iOS and macOS applications was Objective-C. In 2014, Apple introduced Swift as the new official programming language for iOS, iPadOS, macOS, tvOS, and watchOS. The first couple of years saw big changes in the language design. In 2015, the Swift language, supporting libraries, and tools were made open-source. Since the release of Swift 3.0 in 2016 the language stabilized.

With the growing number of available apps relying on evolving third-party libraries, the need for systematic management of dependencies emerged. The events in blue in Fig. 1 show the release dates for the three package managers popular in the iOS ecosystem. A description of these package managers is given in the following.

Package managers

The focus of our research is on libraries that can be used in applications written in Swift, such as iOS or macOS applications. Package managers used in Swift development are CocoaPods, Carthage, and Swift Package Manager (Swift PM).

CocoaPods

CocoaPods (https://cocoapods.org) was released in September 2011 and is the oldest package manager with around 88 thousand libraries. CocoaPods is a centralized package manager. Dependencies are declared in a manifest file called Podfile. When CocoaPods is executed it downloads and compiles all libraries declared in the Podfile. It generates a new Xcode Workspace that has all libraries included. This makes CocoaPods very easy to use, as there is no additional manual work needed.

Information on all libraries and library versions is uploaded to the central Spec repository (https://github.com/CocoaPods/Specs). This means that it is possible to extract information for all packages that have ever been available through CocoaPods. CocoaPods matches libraries through the library name. Each Spec file stored on the central Spec repository contains information on the library such as: library name, repository address, and library version. The library name used within CocoaPods does not need to necessarily match the actual project name.

CocoaPods dependencies are declared in the Podfile simply as follows.

pod <libraryName> <optional versionRequirement>

Resolved dependencies are listed in Podfile.lock under “PODS:” and are given in the following format:

PODS:

- <libraryName> (<exactVersion>)

   - <transitiveDependency> (= <version>)

   - <transitiveDependency> (<exactVersion>)

DEPENDENCIES:

- <libraryName> (>= <version>)

Carthage

The package manager Carthage (https://github.com/Carthage/Carthage) was released in November 2014 as a counterweight to the more heavyweight CocoaPods. Carthage is a decentralized package manager that does not rely on an official central repository of libraries. Libraries can be included through Carthage by simply adding a repository address of a library to the Cartfile. Carthage downloads and compiles these libraries but does not automatically include them in the app projects. Manual work of adding the library to the app project is still needed. This makes using Carthage slightly more complicated than CocoaPods, but it is also a lot more lightweight and developers are not forced to use a generated app project.

Carthage manages about 4.5 thousand libraries (Libraries.io, 2022). This number, however, is an estimate as no official list of repositories exists for this decentralized package manager. Libraries.io (2022) provides a list of 4,498 Carthage libraries that are extracted from Cartfiles hosted on GitHub.

Carthage dependencies are declared in a Cartfile. The dependencies are specified by giving the type of source, name of library and the version requirement, for example:

github "<userName/projectName>" "<version>"

  git "<repoAddress>" >= "<version>"

The manifest file Cartfile.resolved has a very similar format, but contains resolved version numbers instead of version constraints.

Swift PM

The Swift Package Manager (Swift PM) (https://www.swift.org/package-manager/) was released in December 2017. This release date is based on the date of the first release in the Swift PM GitHub repository. Swift PM is the official package manager created by Apple. Swift PM is a decentralized package manager like Carthage. Differently to the other two package managers, Swift PM can also be used to create Swift packages that can be both libraries or applications. This means that Swift PM can be used for example to create a new command line application. Support for iOS applications was not added to Swift PM until 2019 (Elliott, 2020). Since 2019 it is also possible to use Swift PM directly through Xcode, the main IDE for iOS and macOS development.

While Swift PM has no official centralized list of repositories, there are multiple repositories containing information on Swift PM libraries. Libraries.io contains 4,207 libraries, swiftpackageregisty (https://swiftpackageregistry.com) contains 4,348 libraries, and Swiftpack.co (https://swiftpack.co/) contains 12,143 Swift packages. Packages on Swiftpack.co, however, do not seem to be all libraries.

Swift PM dependencies are declared in Package.swift files where the project is declared as an instance of the Package object. The resolution file Package.resolved is a JSON file containing information on all resolved dependencies, both direct and transitive:

{

   "pins" : [

    {

     "identity" : "<libraryName>",

     "kind" : "remoteSourceControl",

     "location" : "<repoAddress>",

     "state" : {

       "revision" : "<revisionHash>",

       "version" : "<exactVersion>"

      }

    }

   ]

}

Exact versions of dependencies can be extracted from the Package.resolved file.

Package manager comparison

Similarities and differences between the three package managers are summarized in Table 1.

Table 1 Similarities and differences between package managers.

Package manager	Centralised	Library inclusion	Changes in workflow	Xcode integration	
CocoaPods	Yes	In workspace	Need to use work-space	Partial integration	
Carthage	No	No automatic inclusion	No changes after manually including libraries	No integration	
Swift PM	No	Automatically included	No changes	Completely integrated	

Dataset of the swift library dependency network

In this subsection, we describe the Swift LDN dataset that we analyze to answer our research questions. The dataset contains information on open source libraries that are available through CocoaPods, Carthage and Swift PM. The dataset consists of libraries, dependencies between libraries, and openly reported vulnerabilities for these libraries. The dataset has been made available on Zenodo (Rahkema & Pfahl, 2022b). We have described how we built the dataset previously in Rahkema & Pfahl (2022a).

The dataset is constructed as a graph database using neo4j (https://neo4j.com/), which stores data as nodes and relationships between nodes. The database contains the following nodes relevant to this study: Project (project including repository information)

App (an analyzed project version)

Library (library version that was referenced from a resolution file of a library)

LibraryDependency (library version that was referenced from a manifest file of a library)

Nodes are connected via the following relationships: (Project)-[:HAS_APP]->(App)

(App)-[:IS]->(Library)

(App)-[:DEPENDS_ON]->(Library)

(App)-[:DEPENDS_ON]->(LibraryDefinition)

A neo4j database was chosen due to its support for large graphs and it has been used previously to create dependency graphs, for example by Benelallam et al. (2019).

The dataset was constructed in three steps as presented in Fig. 2. In the first step, we identified an initial set of libraries to analyze for each package manager. Libraries for CocoaPods were extracted from the CocoaPods central Spec repository. Libraries for Carthage and Swift PM were queried from the libraries.io dataset.

Figure 2 Dataset creation diagram.

The figure shows how the database was constructed. It contains five types of elements: data sources, automatic actions where three different tools were used and lastly actions that were performed manually. First repository URLs were queried and library source code was downloaded for each library, then the dependencies declared through package managers were extracted. In a third step the gathered datasets for each package manager were combined and a snowballing approach was conducted to find any libraries that were referenced but had not been analysed yet.

For CocoaPods, repository URLs were extracted from the official CocoaPods Spec repository. After filtering out erroneous values 73,321 repository addresses remained. For Carthage and Swift PM we queried repository URLs from the libraries.io dataset, where the project platform was either “Carthage” or “SwiftPM”. The query found 3,880 URLs for Carthage and 4,207 URLs for Swift PM.

After the initial set of libraries was extracted, we analyzed the dependencies of these libraries by parsing the manifest and resolution files of each library version. This analysis was done using the three tools: the LibraryDependencyAnalysis (https://github.com/kristiinara/LibraryDependencyAnalysis) shell script, the GraphfiyEvolution (https://github.com/kristiinara/graphifyevolution) source-code analysis tool, and the dependency analysis tool SwiftDependencyChecker (https://github.com/kristiinara/swiftdependencychecker). We analyzed the initial set of libraries for CocoaPods, Carthage and Swift PM. In total 3,094 Carthage, 2,118 Swift PM and 56,822 CocoaPods libraries were successfully analyzed.

After the initial set of libraries was analyzed, the three resulting databases were merged together. The merged database contained 60,084 successfully analyzed libraries with 4,728 library dependencies (of which 1,047 were not analyzed).

The initial list of libraries for Carthage and Swift PM was from 2020. To include more recent libraries as well, we queried all library dependencies from our dataset that had not been analyzed yet. We then tried to analyze each of these libraries. Libraries that are not open source still failed to analyze, but this process succeeded in analyzing newer libraries that are referenced but not yet analyzed. This snowballing process added an additional 451 libraries. Of the initial list of library dependencies that were not analyzed, 311 were not analyzable and for 274 libraries the library name did not match the correct “projectname/username” format.

We give a more detailed description of the dataset in our technical report (Rahkema & Pfahl, 2022c). The final database contains data on 60,533 libraries, 572,131 library versions, and 23,419 dependencies between libraries.

Method

In this section, we first present the research questions that guided our study. Then we describe what analyses we conducted on our dataset to answer each of the research questions. The Jupyter notebook containing analysis scripts can be found on GitHub (https://github.com/kristiinara/LibraryDependencyAnalysis/blob/main/DataAnalysis/notebook.ipynb).

Research questions

The use of package managers to handle the inclusion of third-party libraries and to manage library versions and cascading dependencies has become an important concept and best practice in software development. For this purpose, the following three package managers are commonly used in the Swift ecosystem: CocoaPods, Carthage and Swift PM. In this study, we investigated the following research questions (RQs) to better understand the Swift ecosystem and the role that the three package managers play. Furthermore, we also investigated how the introduction of the newest and official package manager for Swift might have affected the Swift ecosystem. RQ1: How has the popularity of package managers evolved over time?

RQ2: Are CocoaPods, Carthage and Swift PM used concurrently?

RQ3: How does the introduction of new package managers influence the evolution of package manager ecosystems? – RQ3.1: Are existing libraries switching to the newest package manager?

– RQ3.2: Do new libraries prefer the newest package manager?

In the following, we describe the motivation for each research question.

RQ1: How has the popularity of package managers evolved over time?

Given the fast growing number of iOS and macOS applications as well as related software development projects for over more than 10 years, we are interested in how the overall popularity of package managers has evolved in this time-frame. Our assumption is that over time a larger percentage of developers started using package managers for their projects.

RQ2: Are CocoaPods, Carthage and Swift PM used concurrently?

After analysing the popularity of the package managers, we question if the three package managers are used concurrently. Our assumption is that most popular libraries are available through more than one package manager and are therefore also using multiple package managers. For an average library, however, using multiple package managers should not be necessary.

RQ3: How does the introduction of new package managers influence the evolution of package manager ecosystems?

The first package manager in the Swift ecosystem, CocoaPods, was released in 2011, followed by two more package managers in 2014 and 2017. We ask how the introduction of new package managers influences the usage of all available package managers and the evolution of the related ecosystems.

It is probable that new package managers were released because of a lack of desired features provided by the existing package managers. Hence, we see two possible evolution patterns: Each new package manager brings new users that adopt using third party libraries in their applications.

New package managers introduce new features and take over users from existing package managers.

To better understand the evolution of the package managers ecosystems, we first ask if existing libraries are switching to the newest package manager and then ask if new libraries prefer the newest package manager. Given that a migration between package managers might be costly, we assume that there is little migration between package managers, but that more and more new libraries would prefer the newest package manager.

Data analysis

In this section we describe our approach used to answer the three research questions.

RQ1: How has the popularity of package managers evolved over time?

For the years 2012 to 2021, we count the number of unique libraries that have declared a dependency through CocoaPods, Carthage, Swift PM, or that were using no package manager at all. For each year we find the newest version of each library and group unique libraries by their use of package managers. If a library uses multiple package managers then it is counted under each of the used package managers.

We calculate and report the percentage of libraries using CocoaPods, Carthage, Swift PM, and no package manager for each year.

RQ2: Are CocoaPods, Carthage and Swift PM used concurrently?

Libraries can belong to a package manager in two different ways. They can either have dependencies or dependents through a package manager. If a library has dependencies through a package manager then the developers of that library actively use this package manager. If a library has dependents through a package manager then developers of other libraries include this library as a dependency through the given package manager.

For all libraries that have dependencies we count the number of libraries that include dependencies through each package manager and each combination of package managers. For all libraries that have dependents we count the number of libraries that are included as dependencies through each package manager and each combination of package managers. Library dependencies through different package managers are made comparable by using the library repository address as the unique characteristic.

The percentages of all these combinations are calculated for four different years: 2016, 2018, 2020 and 2021. Each yearly snapshot is derived by only including libraries, that have had updates in the given year and taking into account the last version of that library within the year.

RQ3.1: Are existing libraries switching to the newest package manager?

We analyse how libraries migrate between package managers by recording yearly changes of package manager use and drawing a Sankey diagram. For this, we group libraries by year and package manager. For each year, library, and package manager we then record if the library used the same package manager in the previous year. If not, we record if the library used a different package manager in the previous year. If yes, we count this as a migration between package managers.

We then plot a Sankey diagram displaying the migrations between package managers and between using a package manager and not using a package manager. We also plot the percentage of libraries migrating from each of the package managers to using no package manager or other package managers.

RQ3.2: Do new libraries prefer the newest package manager?

Based on the analysis for RQ2.2 we report how many new libraries use each package manager for each year.

Results

In the following, we present the answers to our research questions.

RQ1: How has the popularity of package managers evolved over time?

We analyzed how many libraries are using a package manager vs. not using a package manager. Table 2 shows that the percentage of libraries using no package manager has steadily decreased from 97.8% in 2012 to 84.3% in 2021. The number of libraries presented in Table 2 for each year is calculated by finding libraries that were updated at least once during the given year.

Table 2 Percentage of libraries using each package manager for years 2012 to 2021.

Year	None	CocoaPods	Carthage	Swift PM	# of Libraries	
2012	97.8	2.7			1,067	
2013	95.6	6.1			3,085	
2014	93.6	8.5	0.5		5,837	
2015	92.5	7.9	3.4		9,920	
2016	91.6	7.3	5.5		15,068	
2017	90.3	7.7	5.4	0.9	16,432	
2018	88.5	8.0	5.1	2.2	16,523	
2019	87.4	8.6	5.2	4.3	15,668	
2020	86.3	8.4	4.7	6.7	12,667	
2021	84.3	8.0	4.6	9.0	9,504	

The overall number of actively maintained libraries in the Swift ecosystem grew up to 2018 and has been falling since.

Most libraries in the Swift ecosystem do not use a package manager. The percentage of libraries using a package manager has increased steadily (from 2.2% in 2012 to 15.7% in 2021).

RQ2: Are CocoaPods, Carthage and Swift PM used concurrently?

For all analyzed libraries we counted how many libraries used no package managers, one package manager or multiple package managers. We took into account the last version of each library, in total 60,527 library versions.

We found 52,869 (87.3%) libraries not using any package manager. Of the 7,540 libraries that had dependencies, 4,718 (62.6%) libraries used only CocoaPods, 1,141 (15.1%) libraries used only Carthage, and 1,001 (13.3%) libraries used only Swift PM. In total, thus, 6,860 (91.0%) of libraries with dependencies only use one package manager. The remaining 680 (9%) libraries use multiple package managers, divided between Carthage and Swift PM in combination with 352 (4.7%) libraries, Carthage and CocoaPods with 162 (2.1%) libraries, CocoaPods and Swift PM with 126 (1.7%) libraries, and 41 (0.5%) libraries that use all three package managers.

We also calculated these numbers for four snapshots for the years 2016, 2018, 2020 and 2021. Years 2016 and 2018 were considered to capture the change in the LDNs after the introduction of Swift PM in 2017. Years 2020 and 2021 were considered to see the current trends in the LDNs. The snapshots were constructed by only considering the last version of a library for each year. If a library did not have any versions released during a specific year it was not counted. The top row in Fig. 3 shows how the concurrent use of the three package managers has evolved. In 2016, 63% of libraries with dependencies used CocoaPods and 40% of libraries used the Carthage package manager, with a 3% overlap between these two package managers. After Swift PM was introduced in 2017, more and more libraries started using it. In the following years, 370 (14.5%), 857 (38.2%) and 857 (46.6%) of libraries used Swift PM in 2018, 2020 and 2021 respectively. Multiple package managers were concurrently used by 62 (3%), 258 (11%), 377 (17%) and 317 (17%) libraries in 2016, 2018, 2020 and 2021 respectively.

Figure 3 Evolution of the concurrent use of package managers (four snapshots).

Four snapshots of Venn diagrams show how the use of the three package managers has changed over time. Concurrent use of package managers can be seen in each snapshot. Most libraries, however, use and are referenced by only one package manager.

For the analysis of how libraries are included as a dependency, we analyzed all dependencies between libraries and counted the number of libraries that are included through each package manager. For each library, we took into account the last version of the dependent library. In total there were 3,891 libraries with dependents. We found that 2,410 libraries (61.9%) were only used through CocoaPods. Additionally, 562 (14.4%) and 469 (12.0%) libraries were only used through Carthage and Swift PM respectively. The remaining 450 (11.6%) libraries were included through multiple package managers. This number comprises of the following usages: 121 (3.1%) libraries through all three package managers, 120 (3.1%) libraries through Carthage and CocoaPods, 103 (2.6%) libraries through Carthage and Swift PM and 79 (2.0%) libraries through CocoaPods and Swift PM. Overall 2,730 (70.2%) of libraries were used through CocoaPods, 906 (23.3%) of libraries were used through Carthage and 799 (20.5%) of libraries were used through Swift PM.

We also analyzed the dependencies between libraries for the four snapshots for 2016, 2018, 2020 and 2021. The snapshots were calculated by only considering dependents that had a released version in the given year. The last version of the dependent in each year was taken into account. For each library that had dependents in the given year, we counted how many of these dependents were declared through each package manager. The distribution of libraries for these four different snapshots can be seen in bottom rows of Fig. 3. In 2016 721 (68.7%) of libraries were referenced through CocoaPods, 415 (39.6%) of libraries were referenced through Carthage and 87 (8.3%) of libraries were referenced through both package managers. After Swift PM was introduced in 2017 the percentage of libraries referenced through Swift PM grew to 207 (14.5%) in 2018, 489 (33.1%) in 2020 and 508 (38.6%) in 2021. At the same time the number of libraries included through Carthage shrunk from 415 (39.6%) in 2016 to 288 (21.9%) in 2021.

All package managers are used concurrently with some overlaps. A total of 9% of all libraries use multiple package managers and 11.5% are referenced through multiple package managers. However, most libraries are related to one package manager only. A total of 62.6% of the libraries used only CocoaPods, 15.1% Carthage, and 13.3% Swift PM. Of all libraries with dependents, 61.9% were only referenced through CocoaPods, 14.4% only through Carthage, and 12.0% only through Swift PM.

RQ3: How does the introduction of new package managers influence the evolution of package manager ecosystems?

Carthage and Swift PM were introduced after CocoaPods. We analysed how the introduction of new package managers has influenced the evolution of the Swift package manager ecosystem by studying how libraries migrate between package managers and if new libraries prefer the newest package managers.

Figures 4 and 5 provide an overview of the evolution of the package managers. For both figures the width of the connecting lines is proportional to the number of libraries that migrated between the given package managers. Figure 4 shows the evolution for all libraries. We see, as discussed earlier, that most libraries do not use a package manager. There are, however, migrations of libraries between package managers and, furthermore, libraries from not using a package manager to using a package manager and the other way around.

Figure 4 Sankey diagram for all libraries.

The Sankey diagram shows how libraries migrate between package managers. Gray lines show libraries that keep using the same package manager, orange lines show libraries migrating between package managers and green lines show new libraries that started using a package manager.

Figure 5 Sankey diagram for libraries that use a package manager.

The Sankey diagram shows how libraries migrate between package managers. Gray lines show libraries that keep using the same package manager, orange lines show libraries migrating between package managers and green lines show new libraries that started using a package manager. Only libraries that at some point during its lifetime use a package manager are included in the Figure.

Figure 5 zooms into the same picture by discarding libraries that neither use a package manager nor participate in migrations between using a package manager and not using a package manager. We can see that there are rather large migrations from Carthage to Swift PM between the years 2018 and 2021. In the following sections we analyse these migrations in more detail.

RQ3.1: Are existing libraries switching to the newest package manager?

We analysed the number of libraries that migrate to other package managers for each of the package managers. Figure 6 shows the migrations of package manager users from CocoaPods to not using a package manager (labeled as “to None”) and to using newer package managers, i.e., Cartage (since 2014) and Swift PM (since 2017). While most libraries keep using CocoaPods over time, the data suggests that there is a growing trend to migrate from CocoaPods to not using a package manager at all until 2016. The appearance of Carthage in 2015 seems to stop and even revert this trend until it stabilizes from 2018 onward. In addition, Carthage manages to attract libraries from CocoaPods during the first years after its introduction. However, when Swift PM appears in 2018, the attraction of Carthage for libraries managed by CocoaPods seems to stall and is replaced by that of Swift PM. Moreover, the percentage of libraries migrating from CocaPods to Swift PM seems to be slowly growing.

Figure 6 Percentage of libraries migrating from CocoaPods.

The bars show the percentage of libraries using different package managers that migrated from CocoaPods in each given year. The red part of the bar shows percentage of libraries that kept using CocoaPods this year. Gray part of the bar shows libraries that stopped using a package manager all-together. The blue and yellow part of the bar shows libraries that migrated to Carthage and Swift PM, respectively.

Figure 7 shows migrations from Carthage. In contrast to CocoaPods, there is a large percentage of libraries migrating away from Carthage from the beginning. This is particularly visible in 2015, the first year after Carthage was introduced. Almost one third migrates to CocoaPods and another third stops using a package manager. After the release of Swift PM, the largest migration is towards Swift PM with an increasing growth pattern.

Figure 7 Percentage of libraries migrating from Carthage.

The bars show the percentage of libraries using different package managers that migrated from Carthage in each given year. The blue part of the bar shows percentage of libraries that kept using Carthage this year. Gray part of the bar shows libraries that stopped using a package manager all-together. The red and yellow part of the bar shows libraries that migrated to CocoaPods and Swift PM, respectively.

Figure 8 shows the migrations from Swift PM. There are small migrations towards Carthage and CocoaPods. The majority of the libraries using Swift PM, however, continues using Swift PM.

Figure 8 Percentage of new libraries using each of the package managers.

The bars show the percentage of libraries using each of the three package managers each year. Colors red, blue, and yellow are used for CocoaPods, Carthage, and Swift PM, respectively. Libraries using multiple package managers are not included.

Most libraries that use a package manager keep using CocoaPods. However, given the large portion of libraries not using a package manager at all, there seems to be room for an alternative to CocoaPods. Although Carthage gained some attraction after its introduction, the more successful new package manager seems to be Swift PM as it does not only attract Carthage users but also seems to have little losses.

RQ3.2: Do new libraries prefer the newest package manager?

We analyzed which package managers are used by new libraries. Figure 9 shows how the percentage of libraries using each package manager has changed over time. After Carthage was released in 2014 the percentage of libraries using CocoaPods has stayed between 50% and 70%. Most popular years among new libraries for Carthage were 2015, 2016, and 2017. After the release of Swift PM in 2017 its popularity among new libraries has steadily increased.

Figure 9 Percentage of libraries migrating from Swift PM.

The bars show the percentage of libraries using different package managers that migrated from Swift PM in each given year. The yellow part of the bar shows percentage of libraries that kept using Swift PM this year. Gray part of the bar shows libraries that stopped using a package manager all-together. The red and blue part of the bar shows libraries that migrated to CocoaPods and Carthage, respectively.

CocoaPods keeps its position as the most popular package manager to choose for new libraries. However, since its release in 2017, Swift PM, is growing in popularity. The appearance of Carthage seems to have been a temporary alternative to the more complex CocoaPods.

Discussion

In the following, we discuss the results of the three research questions.

RQ1: How has the popularity of package managers evolved over time?

We saw that the percentage of libraries not using a package manager decreased from 97.8% in 2012 to 84.3% in 2021. We compared this trend to package managers from other ecosystems. Therefore, we additionally calculated the percentage of libraries with dependencies over time for 10 other package managers with sufficient dependency data in Libraries.io (2022): Maven, Packagist, NPM, CPAN, Hex, NuGet, Pub, Puppet, PyPI, and Rubygems. We found that for all 10 package managers the percentage of libraries with dependencies grew over time. Different from the Swift ecosystem, however, the general trend is significantly steeper with most libraries using package managers to declare dependencies by 2020. The only package managers with a slightly similar trend to the Swift ecosystem are Maven and PyPI with 43.1% and 28.5% of libraries declaring dependencies through a package manager for Maven and PyPI respectively. However, libraries in these package managers still declare dependencies more often than in the Swift ecosystem. This might be because developers in the Swift ecosystem seem rather conservative in declaring dependencies, a sentiment that can also be observed in developer forums (Kutjelul, 2022). This sentiment has developed in big part due to the instability of Swift when the language was young. The Swift syntax changed frequently with new versions, making it difficult to migrate from one library version to another. It was also not possible to stay with old versions as Apple requires the use of the latest Swift version to be able to release applications on the App Store. Additionally, Swift comes with fairly rich system libraries, compared with other ecosystems, such as the very lightweight JavaScript.

To ensure that the small number of dependencies is not due to small libraries that are not used by other developers we performed a small check on especially popular libraries. We searched for the best iOS libraries and went through the libraries in the first found list of libraries (Antino, 2024). We matched each of these libraries to libraries in our dataset through the repository URL. We then queried the number of dependencies from our dataset. The results are listed in Table 3. We were not able to match two of the ten libraries. Other libraries confirm our previous results.

Table 3 Number of dependencies for top 10 libaries.

Library name	Number of dependencies	Number of GitHub stars	
RxSwift	N/A	24 k	
Kingfisher	1	22.9 k	
SwiftyBeaver	0	5.9 k	
Nimble	1	4.8 k	
Realm	N/A	16.2 k	
Snapkit	0	19.8 k	
Eureka	0	11.8 k	
Spring	0	14.1 k	
Starscream	3	8.2 k	
CocoaLumberjack	1	13.1 k	

The overall number of libraries in the combined Swift ecosystem has surprisingly decreased since 2018. Swift PM is the only package manager where the absolute number of libraries has increased over this time, indicating an increased adaption by new libraries. The decrease in the number of libraries for CocoaPods has a disproportionate effect on the overall number of libraries due to the difference in how libraries are discovered for the three package managers. The CocoaPods central repository allows the inclusion of all libraries available through the package manager, including small and unused libraries. Libraries for Carthage and Swift PM on the other hand have no central repository. The increase in the adaption of Swift PM could be attributed to efforts by Apple, such as integrating Swift PM directly into Xcode and significantly facilitating the adaption of the package manager.

RQ2: Are CocoaPods, Carthage and Swift PM Used Concurrently?

Three package managers are used in the Swift ecosystem: CocoaPods, Carthage and Swift PM. We expected that the LDNs of these ecosystems overlap, but that there are also libraries that are available only through CocoaPods, Carthage or Swift PM respectively. This assumption proved to be true.

Another, more silent, assumption was that CocoaPods is the largest package manager. The assumption was based on the number of libraries reported by different sources, claiming the number of libraries served by CocoaPods to be around 89,000 and the number of libraries served by Carthage and Swift PM to be around 4,500 each. Our analysis showed that although this is true, the difference between the package managers is not as big as assumed. In 2016 63% of libraries with dependencies used CocoaPods and 40% of libraries used Carthage. In 2021 48% of libraries with dependencies used CocoaPods and 47% of libraries used Swift PM.

An explanation for the smaller difference is that, while CocoaPods has an official central repository, Carthage and Swift PM do not. Therefore, it is not possible to gather all libraries served through Carthage and Swift PM. At the same time the official CocoaPods repository has many incorrect references to libraries. When looking at libraries that either use or are referenced through a package manager the difference between package manager sizes is significantly smaller.

This is an interesting insight for developers who might choose CocoaPods with the expectation of it providing access to 10 times more libraries. We saw that this expectation might not hold true for libraries that are referenced and popular.

RQ3: How does the introduction of new package managers influence the evolution of package manager ecosystems?

Over time the popularity of CocoaPods remains stable. Some libraries switch from CocoaPods to other package managers, but the percentage of these libraries is relatively low. Many libraries, on the other hand, switch away from Carthage. Over time more projects switch from Carthage to Swift PM than from Carthage to CocoaPods, which might be due to the underlying similarity of Carthage and Swift PM. Additionally, Swift PM is integrated into Xcode, making it very easy for developers to use.

In conclusion, the introduction of a new package manager does not necessarily make libraries switch to the newest package manager. The difference in which features are supported by a package manager have an effect on if library developers switch between package managers. Unique features of a package manager can provide stable popularity among developers. If Apple wants to bring more libraries to Swift PM, it might be beneficial to add some features that only exist for CocoaPods so far, for example, a centralized repository (or perhaps a repository for officially vetted libraries).

Summary

While our observations are not sufficient to give a conclusive answer to what makes a package manager attractive, our answers to RQ1, RQ2, and RQ3, shed light on what aspects might contribute to making new package manager attractive to developers. While RQ1 and RQ2 provide relevant background on how the dependency ecosystem has evolved over time, RQ3 provides insights on library developer migration between the package managers showing when developers migrated to new package managers and hence which newly developed features might have impacted the decision to migrate. This includes the following aspects: Ease of use: –Integration with the default development environment.

–Integration into the existing workflow.

–Finding libraries to include as dependencies (e.g., a central repository).

–Adding new libraries.

New features that are not available in existing PMs.

Implementation of popular features of existing PMs.

The good adoption of Swift PM can be explained by the package manager fulfilling most of the ease of use categories. It is integrated with the default development environment, it is integrated in the existing workflow, and adding new libraries is made very easy. The only aspects where it lacks in comparison to CocoaPods is a missing central repository for easier finding of new libraries and a smaller number of supported libraries to due CocoaPods being the oldest package manager.

Threats to validity

In this section, we discuss the potential limitations of how our dataset was constructed and analyzed.

Construct validity

We only look at libraries declared through package managers. It might be possible that some projects are using dependencies, but through other means (e.g., by manually downloading them). Our analysis is based on third-party libraries. Additional analyses are needed to confirm if our results can be generalized to all projects written in Swift, including apps.

Internal validity

The Swift LDN dataset includes open-source libraries only. Additionally, some libraries were excluded as the repository contained no tags.

The library dependency data mostly relies on package manager resolution files. Not every library that uses a package manager includes the corresponding resolution files in the repository. For such repositories, the package manager manifest files are parsed, and the dependency requirements are resolved.

Building the dependency graph by only declaring the exact version of a dependency means that transitive dependencies could in practice be resolved differently. When a transitive dependency is resolved at a later date then it is possible that the actual version of the transitive dependency would not match the version in our dataset. The data on the version ranges does, however, exist in the dataset and could be checked as future work.

External validity

We claim that our results hold for all open-source libraries in the LDNs of the Swift ecosystem, i.e., all open-source libraries that are available through CocoaPods, Carthage and Swift PM. For CocoaPods, the official repository that contains information on libraries available through CocoaPods was used. For Carthage and Swift PM, the information on libraries.io was used as the initial set of libraries. To make sure that newer libraries than 2020 are included and that we do not rely solely on libraries.io, snowballing was used to analyze referenced dependencies that were not analyzed in our dataset yet. This additional step should ensure that we also include libraries that should be in the dataset but that did not exist in the initial set of libraries.

We analyzed the LDNs of the Swift ecosystem between September 2011 and December 2021. We make no claims to how the LDNs might evolve in the future. We saw in our analysis that the introduction of a new package manager can disrupt any trends that might have existed before.

All data and all tools are open source and available on public repositories. We tried to describe each manual process as detailed as possible. Therefore, our study should be reproducible.

Related work

Many studies have been conducted analyzing technical lag in library dependencies (Zerouali et al., 2018, 2019; Decan, Mens & Constantinou, 2018a; Salza et al., 2020; Huang et al., 2019; Stringer et al., 2020) and vulnerabilities in different LDNs (Decan, Mens & Constantinou, 2018b; Zerouali et al., 2022; Düsing & Hermann, 2021; Li et al., 2021; Zimmermann et al., 2019; Alfadel et al., 2020, 2023; Prana et al., 2021). There are no studies analyzing multiple package managers in the same ecosystem. However, studies have compared LDNs of different ecosystems. This section summarises the related work that compares LDNs of multiple ecosystems.

Kikas et al. (2017) analyzed the evolution of LDNs of three languages: JavaScript, Ruby, and Rust. They found that for each package manager the number of libraries is growing. Similarly, the number of direct dependencies and total dependencies per project is increasing. The increase was especially concerning for JavaScript, where the average number of total dependencies grew from one per project to almost 60 between 2011 and 2016. Decan, Mens & Claes (2017) analyzed the LDNs of three package managers npm, CRAN and RubyGems. They found that proportionally there are more packages with dependencies in CRAN (70%) than in npm and RubyGems (60%). They also found that on average there are few direct dependencies and a much higher number of transitive dependencies. The median number of transitive dependencies for CRAN was five, for RubyGems 8, and for npm 22.

Decan, Mens & Grosjean (2019) analyzed the evolution of seven package manager LDNs. They defined and calculated three metrics describing the LDN evolution: the Changeability Index, the Re-usability Index, and the P-Impact Index. They used the libraries.io dataset to analyze how these package manager LDNs change over time. They found that the growth of the number of libraries and dependencies depends on the package manager. Some LDNs have grown linearly, as others have grown exponentially. For most package managers 50% of libraries were updated within 2 months and libraries that are referenced by other libraries are updated significantly more often than libraries, that are not referenced by other libraries. They also found that 26% to 33% of libraries were never updated. They showed that the number of transitive dependencies is significantly higher than the number of direct dependencies. For some of the package managers, the ratio between transitive and direct dependencies is growing. They also pointed out that the average dependency depth is between three and six, depending on the package manager. The libraries.io data set includes partial data about CocoaPods, Carthage, and Swift Package Manager (the three package managers used in iOS development), but this data was incomplete and therefore, these three package managers were excluded from the analysis.

Bogart et al. (2021) analyzed the policies and practices for 18 LDNs. Their analysis showed that ecosystems share values on stability and compatibility, but other values tend to differ. The three top values cited by developers for CocoaPods were quality, stability, and compatibility. Blanthorn, Caine & Navarro-López (2019) used tensor decomposition to study different communities within LDNs. They found big differences between package managers, particularly between Elm and R and the more widespread Python, Java, and JavaScript ecosystems. Korkmaz et al. (2020) found that libraries with a higher number of dependencies tend to have less impact in the LDN.

Kula et al. (2018) analyzed dependency updates in 4,600 Java projects. They found that 81.5% of the studied projects did not update their outdated dependencies. They plotted library usage curves and discovered that new library versions are mostly used by new dependent projects.

Finally, Domínguez-Álvarez, Gorla & Caballero (2022) analyzed the evolution of the CocoaPods library ecosystem by looking at the dominant programming language of libraries. They found that although Swift has gained popularity, most libraries in CocoaPods are still written in Objective-C.

Conclusion

We analysed the Swift LDN to understand what properties make a newly proposed package manager attractive to developers. We first analyzed how the use of the three package managers used in the Swift ecosystem (CocoaPods, Carthage, and Swift PM) has evolved over time. We then analyzed how the package managers are used concurrently and, lastly, we analyzed how the introduction of new package managers has influenced the package manager ecosystem.

We found that most libraries in the Swift ecosystem do not use a package manager. However, the percentage of libraries using a package manager has increased over time (from 2.2% in 2012 to 15.7% in 2021). The percentage of libraries using Carthage peaked in 2016 at 5.5%, the percentage of libraries using CocoaPods has been steadily around 8% since 2018. The percentage of libraries using Swift PM, however, has an increasing trend and reached 9% in 2021. Of all libraries using package managers, 9% use multiple package managers concurrently. We found that some libraries are switching to the newest package manager, CocoaPods, however, is keeping most of its users.

Whether libraries switch between package managers is dependent on the features of the package managers involved. Unique features of a package manager can provide stable popularity among developers. The introduction of a new package manager can be successful if it provides enough features that are lacking from existing solutions. If Apple wanted to bring more libraries to Swift PM, it might be beneficial to add some features that only exist for CocoaPods so far, for example, a centralized repository.

Additional Information and Declarations

Competing Interests

Author Contributions

Data Availability

The authors declare that they have no competing interests.

Kristiina Rahkema conceived and designed the experiments, performed the experiments, analyzed the data, performed the computation work, prepared figures and/or tables, authored or reviewed drafts of the article, and approved the final draft.

Dietmar Pfahl conceived and designed the experiments, authored or reviewed drafts of the article, and approved the final draft.

Rudolf Ramler conceived and designed the experiments, authored or reviewed drafts of the article, and approved the final draft.

The following information was supplied regarding data availability:

The data is available at Zenodo: Rahkema, K., & Pfahl, D. (2022). Dependency Networks of Open Source Libraries Available Through CocoaPods, Carthage and Swift PM (1.1.0) [Data set]. 19th International Conference on Mining Software Repositories (MSR ’22), Pittsburgh, PA, USA. Zenodo. https://doi.org/10.5281/zenodo.6641875.

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
