# Peer review of "The impact of new package managers on the library dependency ecosystem"

_PeerJ Computer Science, doi:10.7717/peerj-cs.2617_

## Round 0.1 · original submission · Major Revisions

The reviewers have recognized the novelty and value of your research, as well as the clarity of your writing and the accessibility of your data and manuscripts.
However, they have also identified several areas that need significant improvement to increase the impact and relevance of your work.

As noted by Reviewer 1, the manuscript does not adequately address the fundamental question posed in the Introduction regarding the features that make a newly proposed package manager attractive to developers. While Reviewer 1 suggests a more thorough analysis of the distinguishing features of package managers, I agree with Reviewer 2 that the problem lies more in the details provided than in the data collected and analyzed. As requested by Reviewer 2, please elaborate on these aspects in the revision. This will provide actionable insights and fulfill the potential of your research to impact the development community.

The reviewers noted that the manuscript would benefit from a clearer articulation of the implications of your empirical findings. Expanding on the takeaways and providing a sound rationale for the unexpected results from RQ1 would significantly enhance the contribution of your work to the existing literature.

Another gap identified by the reviewers is a detailed technical description of the package managers studied. Providing such a description will serve as a valuable resource for readers seeking to understand the complexities of package management in the Swift ecosystem. Reviewer 1 has provided specific suggestions for improvement.

As pointed out by Reviewer 2, you should focus on the most widely used libraries in the Swift ecosystem to ensure that your analysis is more relevant and impactful.
Additionally, please align the motivation of your research with the actual analysis presented in the paper to better structure your manuscript and convey the significance of your work.

Finally, the reviewers noted that the practical utility and lessons learned from your analysis are not clearly conveyed in the current version of the manuscript. I encourage you to address this issue by aligning the motivation of your research with the actual analysis presented in the paper, and by focusing on the most widely used libraries in the Swift ecosystem.

These are the blocking points that I consider important for further consideration of the submission for publication. The reviewers have raised other useful suggestions and concerns that you are free to address at your discretion.

Please also address the minor presentation issues highlighted by the reviewers, such as inconsistencies in the naming of macOS/MacOS and the formatting of references, to enhance the overall quality of the manuscript.

I look forward to receiving your revised manuscript and believe that these revisions will significantly strengthen your contribution to the field.

·

Basic reporting

The paper contributes to the understanding of dependency networks across package managers within a specific software ecosystem (i.e., Apple/Swift). It offers a novel perspective on a topic of increasing relevance in software development. The effort by the authors to ensure the accessibility of their data, scripts, and notebooks through public repositories such as GitHub and Zenodo is commendable and enhances the reproducibility of the research. Furthermore, the clarity and eloquence of the writing, accompanied by clear informative figures (in particular, I would like to highlight the beauty of those sankey plots here), this significantly aid in the dissemination of their findings, making complex concepts more accessible to both practitioners and researchers.

Despite these strengths, there are several areas where the paper could be improved to better achieve its objectives and provide more value to its readers. A primary concern is the paper's failure to address the fundamental question posed in line 539 regarding what properties make a newly proposed package manager attractive to developers. The comparison of adoption rates across package managers, while valuable, misses an opportunity to delve into the specific features that influence these decisions. A more thorough analysis of these features could offer actionable insights and fulfill the paper's potential to impact the development community. Consequently, I recommend a revision that includes a deeper investigation into the distinguishing features of package managers that drive developer preference.

Additionally, the paper could benefit from clearer articulation of its empirical findings' implications. Expanding on the takeaways, perhaps by adding motivation early in the paper, would provide readers with a more straightforward understanding of the study's practical applications. This is particularly relevant in light of the surprising results from RQ1, which suggest most libraries do not utilize a package manager. Providing a sound rationale for these unexpected findings would significantly enhance the paper's contribution to existing literature.

The absence of a detailed technical description of the studied package managers is another critical gap. Such detail would not only enrich the paper's academic rigor but also serve as a valuable resource for readers seeking to understand the nuances of package management in the Swift ecosystem. Addressing this would directly respond to the reader's lingering question about the features that contribute to a package manager's popularity in this context.

On a minor note, revising the paper's title to more explicitly reference the Apple/Swift ecosystem could improve its specificity and relevance. Incorporating a comparative table in the Background section would succinctly summarize the key differences between the package managers studied, providing a quick reference that complements the narrative. Additionally, acknowledging prior work on neo4j dependency networks would situate the paper more firmly within the existing body of research (e.g., see Benelallam, Amine, et al. "The maven dependency graph: a temporal graph-based representation of maven central"). Lastly, there is typo in line 237: “to to”.

Experimental design

The following is a list of the major issues that should be addressed in order to improve the experimental design of this paper before publication (in decreasing order of priority):

1. The paper lacks a more deep, technical description of the studied package managers.

2. The title shouldn’t be a question, instead it should specifically refer to the Apple/Swift ecosystem.

3. In the Background section, I suggest adding a comparative table at the end to summarize the key differences between the studied package managers. For example, the table could include the following columns: # libraries, requires compilation, programming language, is open-source, is centralized/decentralized, IDE support, config file format, etc.)

4. The authors build a neo4j dependency network, this has been done before in the literature but there is no mention of it in the related work. For example, see Benelallam, Amine, et al. "The maven dependency graph: a temporal graph-based representation of maven central."

5. I suggest improving the structure by having the discussion corresponding to each research question together with the results for each question. This will make it easier for the reader to understand the consequences derived from their findings.

Validity of the findings

The following is a list of the major issues that should be addressed in order to improve the validity of the findings of this paper before publication (in decreasing order of priority):

1. The key question of “what properties make a newly proposed package manager attractive to developers” (line 539) is really not answered in the paper. Instead, a comparison of adoption across three package managers is performed. It would have been much more interesting to study the features that are common to these package managers, and which are the ones that make them different and preferred by developers.

2. Which are the actionable insights derived from this empirical study? That’s not clear. I suggest expanding the takeaways (e.g., add more motivation in line 47).


3. Results from RQ1 are surprising and counterintuitive. Why that most libraries don’t use a package manager. Please some sound explanation to justify this result.

4. After reading the paper, I ask myself: So, what are the features of a package manager that make them popular in the Swift ecosystem? This fundamental question is not answered anywhere in the paper.

5. I suggest adding concrete examples of libraries, for example to support the analysis of RQ2 regarding libraries shared among distinct package managers.

Additional comments

While the paper makes a valuable contribution to our understanding of package manager dependency networks, addressing the highlighted weaknesses could significantly enhance its impact and relevance to the field.

Cite this review as

Reviewer 2 ·

Basic reporting

The paper focuses on the impact of introducing new package managers in the Swift ecosystem. The authors analyze the influence of Carthage and Swift Package Manager (Swift PM) on the popularity of CocoaPods. They found that while the introduction of these new package managers did not significantly affect CocoaPods' popularity, Carthage users are increasingly migrating to Swift PM. This trend could be due to the fundamental differences between CocoaPods and the other two package managers, as well as the similarities between Carthage and Swift PM. The authors speculate that Apple could increase the popularity of Swift PM by adding features that have so far only been available in CocoaPods, such as a central repository. The paper emphasizes the importance of understanding the dynamics of library dependency networks in software development ecosystems. It also highlights the potential impact of introducing new tools and technologies in these ecosystems.

The paper uses clear, unambiguous, professional English language.

The introduction and background offer enough context to understand the domain. The literature is well referenced and relevant.

Figures are relevant, high quality, well labeled and described.

Raw data (and analysis scripts) are supplied, so that reproducibility is possible.

Experimental design

The paper includes original primary research within Scope of the journal.

The research questions are defined in the paper, but their relevance throws some doubts. In addition they show not be meaningful with the aim of the paper of discussing properties and functionalities of a "good" package manager. It is not clear how the research fills an identified knowledge gap -- besides the fact that there is no paper that analyzes three package managers from the same ecosystem.

Rigorous investigation have been performed to a high technical and ethical standard.

Methods are described with sufficient detail and there is enough information to replicate, and --as noted above-- that the raw data and analysis scripts are available.

Validity of the findings

As said, the article is well written, the methodology used is properly detailed, it has a lot of data that is conveniently displayed and analyzed, and a reproduction package is provided.

But at the end of the day, I am left with the impression that what is offered is a very superficial analysis, and that I do not know very well what the authors want to convey with this research. It is not clear to me what the practical usefulness of the analysis itself is, nor what lessons learned can be extracted from it.

The abstract is already confusing and it is not clear what it is intended to achieve. Perhaps using a structured abstract scheme can make it better and help determine what is done and, especially, why it is done.

In the conclusion it is said that "We analyzed the Swift LDN to understand what properties make a newly proposed package manager attractive to developers.", but I have not found anything about this in the article. There is no analysis on the properties that such a new package manager should have; and probably the method for obtaining those differ from the one that has been used in this study.

And the last paragraph of the conclusions "Whether libraries switch between package managers is dependent on the features of the package managers involved. Unique features of a package manager can provide stable popularity among developers. The introduction of a new package manager can be successful if it provides enough features that are lacking from existing solutions. If Apple wanted to bring more libraries to Swift PM, it might be beneficial to add some features that only exist for CocoaPods so far, for example, a centralized repository." is not supported by the rest of the research either. This would require an analysis of the features and maybe some qualitative analysis.

One of the results is that most Swift libraries do not use any package manager. But this may be because they are not popular or rarely used libraries, as it is common in open source development. Perhaps focusing on those libraries that are most widely used makes more sense.

In short, the paper lacks in its current state a motivation that is aligned with what is then analyzed in the paper. In addition, the impact in its current form is limited; the desired outcomes are not the ones offered with the method -- and the results shown are interesting but have, at this point, marginal practical significance.

Additional comments

There are some minor presentation issues that I have written down and that are easy to address:

Consistency: some times it appears as macOS, other times as MacOS

Line 41: The introduction date of Swift PM is presented (2017), but not of the two other. Probably this should be included as well

Line 55: e.g.,Bioconductor -> e.g., Bioconductor

Line 73: explain what Xcode is

Lines 84-89: Section numbers do not appear

Line 151: Libraries.io(Libraries.io, -> Libraries.io (Libraries.io,

Line 192: Zenodo(Rahkema and Pfahl, -> Zenodo (Rahkema and Pfahl,

Line 211 and 212: ”Carthage” or ”SwiftPM” -> ``Carthage” or ``SwiftPM”

Line 382: ẗo None¨) -> ``to None'')

Line 411: libraries.io(Libraries.io, 2022) -> libraries.io (Libraries.io, 2022)

Line 458: If libraries switch between package managers is dependent on the features of the package managers involved. -> rewrite

Line 532: They found, that -> They found that

Line 535: Alvarez et al.(Domı́nguez-Álvarez -> Domínguez-Álvarez et al. (Domı́nguez-Álvarez

Cite this review as

---

## Round 0.2 · Minor Revisions

Thank you for your thorough revision addressing the reviewers' comments. Your changes have improved the manuscript, particularly in terms of technical descriptions and comparative analysis of the package managers.

I share Reviewer 2's perspective that the manuscript could benefit from further depth in its analysis. While deeper analysis could be achieved given enough time and resources, what appears to be more pressing here is a refinement in how the findings are framed. The fundamental question about what makes a new package manager attractive to developers, while partially addressed through your analysis, could benefit from a more explicit synthesis of your insights.

To address this, I recommend the following revisions:

1) Gather the various practical insights currently scattered throughout your Results and Discussion sections into a dedicated concluding paragraph (either in Discussion or Conclusion) or a subsection, and present these recommendations in a structured manner that clearly connects them to your original research question about package manager adoption.

2) Consider adding theoretical recommendations that could guide future package manager development. These might include principles derived from your empirical findings about feature adoption patterns and user migration behaviors.

3) A concrete, stated answer to what makes a package manager attractive may not be fully achievable through your current methodology. While I would appreciate such an attempt, it is not compulsory for acceptance. Consider instead explicitly acknowledging how your findings contribute to understanding this question. Consider framing your insights as contributing factors rather than definitive answers.

Please note that this revision does not require additional data collection or analysis, but rather focuses on better synthesizing and presenting your existing findings.

·

Basic reporting

no comment

Experimental design

no comment

Validity of the findings

no comment

Additional comments

The authors have effectively addressed all my major concerns regarding this paper.

Cite this review as

Reviewer 2 ·

Basic reporting

I thank the authors for their efforts in revising the paper.

While I appreciate the addressed minor issues and clarified purpose, I remain concerned about the depth of the analysis.

The additional analysis in Table 3 does not fully address the concerns raised in the previous review. I had hoped for a more substantial revision that would delve deeper into the subject matter. I concur with the other reviewer's assessment that the analysis lacks depth, and in my opinion this superficiality remains in this version.

Given the requested major revision, I had anticipated more substantial changes to the paper. I had hoped for a more comprehensive revision that would address the fundamental issues raised in the previous review.

Experimental design

Nothing to comment here.

Validity of the findings

Nothing to comment here.

Additional comments

Other, minor issues, easy to fix:

Page 3: A description of the these package -> A description of these package

Page 4: Mac OS -> macOS

Page 6: for example by Benelallam et al. Benelallam et al. (2019). -> polish

What does "Total" mean in Table 2? Would "Number of Packages" be better?

Page 13: Pypi -> PyPI

Page 14: forums (kutjelul, 2022). -> forums (Kutjelul, 2022).

Cite this review as

---

## Round 0.3 · accepted · Accept

Thank you for addressing the comments. I have inspected your response letter as well as the changes, and I am satisfied with them. I think that the submission is now ready for publication.